# Effects of Supplemental Lighting on Flavonoid and Anthocyanin Biosynthesis in Strawberry Flesh Revealed via Metabolome and Transcriptome Co-Analysis

**DOI:** 10.3390/plants13081070

**Published:** 2024-04-10

**Authors:** Shen Chen, Xiaojing Wang, Yu Cheng, Hongsheng Gao, Xuehao Chen

**Affiliations:** 1School of Horticulture and Landscape Architecture, Yangzhou University, Yangzhou 225009, China; chenshen999@126.com (S.C.); yc950730@163.com (Y.C.); 2Key Laboratory of Plant Resource Conservation and Germplasm Innovation in Mountainous Region, Institute of Agro-Bioengineering, Guizhou University, Guiyang 550025, China; xjwang8@gzu.edu.cn

**Keywords:** strawberry fruits, red/blue LED light, UVB, flavonoids and anthocyanins, RNA-seq, metabolism

## Abstract

The spectral composition of light influences the biosynthesis of flavonoids in many plants. However, the detailed composition of flavonoids and anthocyanins and the molecular basis for their biosynthesis in strawberry fruits under two light-quality treatments, red light supplemented with blue light (RB) and ultraviolet B (UVB) irradiation, remain unclear. In this study, the content of flavonoids and anthocyanins was significantly increased in strawberry fruits under RB light and UVB, respectively. The content of flavonoids and anthocyanins in strawberry fruits under UVB light was dramatically higher than that in strawberry fruits irradiated with RB light, and a total of 518 metabolites were detected by means of LC-MS/MS analysis. Among them, 18 phenolic acids, 23 flavonoids, and 8 anthocyanins were differentially accumulated in the strawberry fruits irradiated with red/blue (RB) light compared to 30 phenolic acids, 46 flavonoids, and 9 anthocyanins in fruits irradiated with UVB. The major genes associated with the biosynthesis of flavonoids and anthocyanins, including structural genes and transcription factors (TFs), were differentially expressed in the strawberry fruits under RB and UVB irradiation, as determined through RNA-seq data analysis. A correlation test of transcriptome and metabolite profiling showed that the expression patterns of most genes in the biosynthesis pathway of flavonoids and anthocyanins were closely correlated with the differential accumulation of flavonoids and anthocyanins. Two TFs, bZIP (*FvH4_2g36400*) and AP2 (*FvH4_1g21210*), induced by RB and UVB irradiation, respectively, exhibited similar expression patterns to most structural genes, which were closely correlated with six and eight flavonoids, respectively. These results indicated that these two TFs regulated the biosynthesis of flavonoids and anthocyanins in strawberry fruit under RB light and UVB, respectively. These results provide a systematic and comprehensive understanding of the accumulation of flavonoids and anthocyanins and the molecular basis for their biosynthesis in strawberry fruits under RB light and UVB.

## 1. Introduction

Cultivated strawberry (*Fragaria × ananassa*) is a perennial herb plant that belongs to the Rosaceae family and genus *Fragaria*, and it is a valuable crop cultivated worldwide. Strawberry fruits are rich in fiber, micronutrients, ascorbic acid, flavonoids, and anthocyanins and offer a broad variety of sensory elicitation and health benefits to the consumer [1]. The coloring of fruit is an important phase in its life cycle and results from flavonoids, carotenoids, and anthocyanins [2]. Color and appearance are often the initial criteria used by consumers to assess the quality of fruits and vegetables, which can influence food consumption and purchases [3]. Numerous studies have revealed that light is the most essential environmental factor involved in regulating the biosynthesis of flavonoids and anthocyanins [4,5,6,7,8,9]. However, the molecular mechanism governing the different light-quality-induced biosynthesis of anthocyanins in strawberry fruit remains unclear.

As a major component of plant-specialized metabolism, phenylpropanoid biosynthetic pathways provide anthocyanins for pigmentation and flavonoids such as flavones for protection against various biotic and abiotic stresses [10]. Anthocyanins are a class of water-soluble flavonoids widely present in plants [11]. In plants, anthocyanins play a key role in plant reproduction by attracting pollinators and seed dispersers along with helping plants protect against many abiotic and biotic stresses [12]. Anthocyanin synthesis is not only regulated by internal genetic factors but is also affected by external factors such as light, temperature, and nutritional levels [13]. The biosynthesis of anthocyanins is genetically determined by structural and regulatory genes. Several key genes, such as phenylalanine lyase (PAL), 4-coumarate-CoA ligase (4CL), cinnamic acid hydroxylase (C4H), chalcone isomerase (CHI), chalcone synthase (CHS), flavonoid 3-hydroxylase (F3H), flavonol synthase (FLS), flavonoid 3′,5′-hydroxylase (F3′5′H), dihydroflavonol 4-reductase (DFR), and anthocyanidin synthase (ANS), are involved in different steps of the flavonoid synthesis pathway [14,15]. Anthocyanins are one of the end products of the flavonoid synthesis pathway, which requires further glycosylation by UDP-glucosyltransferase and glutathione S-transferase [16]. Conserved structural genes involved in flavonoid biosynthesis have been identified in many plants, while their regulatory mechanisms vary [17]. Numerous studies have shown that several TF families, including WRKY, Dof, MADS-box, bZIP, MYB, bHLH, WD40, and NAC, have been reported to be involved in regulating the biosynthesis of flavonoids and anthocyanins in many plants [3,18,19,20,21,22,23,24,25].

Anthocyanin biosynthesis is also affected by external environmental factors such as temperature, light, and drought [26,27]. Among them, light is the main environmental factor influencing plant growth and development and regulating the biosynthesis of flavonoids and anthocyanins [4]. The effects of supplemental light quality on the biosynthesis of flavonoids and anthocyanins have been investigated in many plants [5,6,28,29]. Liu et al. (2018) revealed that light quality affects flavonoid production by regulating the related gene expression in *Cyclocarya paliurus* [30]. Blue light could induce the biosynthesis of flavonoids in *Epimedium sagittatum* [31]. RB light increased the flavonoid content in *A. roxburghii* by enhancing the expression of several related genes [32]. Illumination with blue light-emitting diodes (LEDs) induced anthocyanin accumulation in red lettuce [33]. Zhang et al. (2018a) showed that blue and red light increased the contents of total anthocyanins, pelargonidin 3-glucoside, and pelargonidin 3-malonyl glucoside in strawberry fruits [34]. In strawberries, red and red/blue light caused earlier fruit coloration by increasing the total anthocyanin content [35,36]. Moreover, UVB plays an important role in regulating the accumulation of flavonoids and anthocyanins in many plants [37,38,39]. Li et al. (2021) found that UVB could promote flavonoid biosynthesis and improve the fruit quality of blueberries [4]. Tsurunaga et al. (2013) revealed that irradiation with UVB > 300 nm increased the levels of anthocyanin and rutin, as well as the DPPH radical-scavenging activity [40]. UVB-visible irradiation caused the accumulation of anthocyanins and quercetin glycosides in “Gala” and “Royal Gala” apple fruit skin [41]. Li et al. (2021) revealed that a long-term preharvest UVB treatment can increase the anthocyanin content of blueberries in the green and mature fruit stages [4]. Various sensory photoreceptors, such as red- and far-red-light-sensing phytochromes (phys: phyA to phyE), blue/ultraviolet-A photoreceptors (cry1, cry2, and cry-DASH), phototropins (phot1 and phot2), Zeitlupe family members (zeitlupe, flavin-binding kelch repeat F-BOX1, LOV kelch protein 2), and UV resistance locus 8 (UVR8), have been identified in many plants [42,43,44]. Further research found that photoreceptors inhibited the E3 ubiquitin ligase COP1/SPA activity. Elongated hypocotyl 5 (HY5), a direct target of COP1, can activate the structural genes and transcription factors in the flavonoid pathway in response to light [45]. In addition, the phytochrome interacting factors (PIFs) family members have been confirmed to interact with phytochromes and participate in anthocyanin biosynthesis by binding the promoters of related genes [46].

Strawberry, a rich source of phytochemicals (ellagic acid, anthocyanins, quercetin, and catechin) and vitamins (ascorbic acid and folic acid), has been highly ranked among dietary sources of polyphenols and in terms of its antioxidant capacity [47]. Particularly, anthocyanins are natural colorants with health-promoting properties, which have received much attention. Light is one of the most important factors that affect anthocyanin accumulation [48,49]. However, uneven coloration and defective quality due to inadequate anthocyanins often occur in strawberry cultivation with poor light conditions. In this study, we evaluated the effects of a long-term environmentally based red and blue light treatment on strawberries cultivated in a greenhouse. The objective of this study was to clarify the effects of red and blue light on strawberry fruit development and maturation. We focused specifically on the molecular regulatory network underlying the red/blue-light-triggered anthocyanin metabolism, which revealed the stage-specific red/blue-light-responsive fruit-coloration mechanism and a novel regulatory network fine-tuning anthocyanin accumulation in strawberry fruits.

## 2. Results

### 2.1. Morphological Traits and Physiological Indicators under Red/Blue Light (RB) and UVB Light

As shown in Figure 1, compared to strawberry fruits under natural light, strawberry fruits under supplementary RB and UVB light showed accelerated color transition during ripening (Figure 1A,B). The flavonoid contents were significantly increased in strawberry fruits under RB and UVB light treatments (385.28 µg/g and 406.59 µg/g, respectively) compared to the control group (57.68 µg/g, Figure 1C). The anthocyanin contents in strawberry fruits irradiated with RB and UVB lights were 1.15 and 1.67 times that of the control group, respectively (Figure 1C). The anthocyanin contents in strawberry fruits under UVB light were dramatically higher than in strawberry fruits irradiated with RB light. The lignin content was slightly increased in strawberry fruits treated with RB light (17.78 µg/g) compared to the control group (15.49 µg/g), while UVB light can slightly reduce the lignin content (14.87 µg/g) (Figure 1D). There was no difference in the contents of TA and total sugars between the control group and the experimental group (RB and UVB light) (Figure 1E). In addition, the activities of antioxidant enzymes were further investigated. As shown in Figure 1F–I, the contents of POD, CAT, SOD, and MDA were significantly increased in strawberry fruits under RB light treatments (12.76 U/g, 42.99 U/g, 6.74 U/g and 36.59 nmol/g, respectively) compared to the control group (9.83 U/g, 21.69 U/g, 6.03 U/g, and 30.46 nmol/g, respectively) The contents of POD, CAT, SOD, and MDA strawberry fruits irradiated with UVB lights were 1.38, 4.14, 1.45, and 1.37 times those of the control group, respectively. The activities of POD, CAT, SOD, and MDA in strawberry fruits irradiated with UVB light were higher than those in strawberry fruits under RB light.

### 2.2. Metabolome Profile of Strawberry Fruits under RB Light and UVB Light

A total of 518 metabolites were identified in strawberry fruits under RB light and UVB light (Appendix A) and were divided into four classes: flavonoids, phenolic acids, lignins and coumarins, and tannins. In PCA, PC1 and PC2 explained 31.1% and 23.2% of the variation, respectively. PC1 separated the nine samples by the different light treatments, and PC2 separated samples by the different repetitions (Figure 2A). Flavonoids (including 245 metabolites) were the most abundant, followed by phenolic acids (208), tannins (58), and lignins and coumarins (7). Metabolites with VIP > 1 and |FC| ≥ 2 were defined as differentially accumulated metabolites (DAMs). A comparison between RB and CK showed that 10 metabolites increased and 35 decreased. The increased metabolites included 4 flavonols, 2 anthocyanins, 2 phenolic acids, and 2 coumarins, and the decreased metabolites included 16 phenolic acids, 5 flavones, 4 dihydroflavonoids, 4 flavonols, 2 flavanols, 2 anthocyanins, 1 chalcone, and 1 lignin. A comparison between UVB and CK showed that 19 (5 phenolic acids, 3 flavones, 3 dihydroflavonoids, 7 anthocyanins, and 1 flavonol) and 65 (25 phenolic acids, 15 flavonols, 8 flavones, 7 coumarins, 5 dihydroflavonoids, 2 anthocyanins, 2 flavanols, and 1 chalcone) metabolites were up-regulated and down-regulated, respectively. The number of differentially expressed metabolites between RB and UVB was 88, of which 53 (23 phenolic acids, 11 flavones, 7 coumarins, 6 flavonols, 3 dihydroflavonoids, 1 flavanol, 1 anthocyanin, and 1 chalcone) and 35 (17 phenolic acids, 5 dihydroflavonoids, 5 flavones, 4 flavonols, 2 coumarins, 1 anthocyanin, and 1 flavanol) metabolites were increased and decreased, respectively. Notably, 23, 31, and 10 differentially expressed metabolites existed in RB vs. UVB and RB vs. CK, RB vs. UVB and UVB vs. CK, and RB vs. CK and UVB vs. CK, respectively. Four flavones, including liquiritigenin-4′-O-Glucoside, phloretin-4′-O-(6″-p-Coumaroyl)glucoside, pinocembrin-7-O-(2″-O-arabinosyl)glucoside, and pinocembrin-7-O-(6″-O-malonyl)-glucoside, were present in all three comparison groups (Figure 2B). Further analysis found that RB light and UVB light mainly induced the accumulation of four flavonoids (Phloretin-4′-O-(6″-p-Coumaroyl)glucoside, Isovitexin-7-O-(6″-sinapoyl)glucoside, vitexin-2″-O-rhamnoside, and Malvidin-3-O-glucoside) in strawberry fruits, indicating that these flavonoids could be involved in RB/UVB-induced strawberry fruit coloration. In addition, there were two anthocyanins (Cyanidin-3-O-sambubioside and Cyanidin-3-O-arabinoside) and six anthocyanins (Cyanidin-3,5-O-diglucoside, Delphinidin-3-O-rutinoside-5-O-glucoside, Pelargonidin-3-O-sophoroside, Pelargonidin-3-O-(6-O-malonyl-beta-D-glucoside), Pelargonidin-3,5-O-diglucoside, and Pelargonidin-3-O-galactoside), suggesting that these anthocyanins may contribute to the fruit coloration.

The identified DAMs from each comparison group were further annotated using the KEGG database. As shown in Figure 3, the DAMs from three comparison groups were mainly enriched in the biosynthesis of secondary metabolites (ko00941), flavone and flavonol biosynthesis (ko00944), and anthocyanin biosynthesis (ko00942) (Figure 3 and Appendix A).

### 2.3. Transcriptome Analysis of Strawberry Fruits Treated with RB Light and UVB Light

Transcriptome sequencing was conducted on nine samples (each with three biological replicates) of strawberry fruits at the red fruit stage after treatment with RB and UVB light. In the present study, 179.07 Gb of clean data was obtained. The clean data for each sample reached about 5.77 Gb, and the Q30 base percentage was more than 91.82% (Appendix A). The correlation coefficients of normalized gene expression between any two biological replicates for all samples were greater than 0.96 (Figure 4A), indicating that the biological replicates were very good, and the data were used to detect the DEGs. Genes with an absolute fold change of >2 and FDR value of <0.05 were considered DEGs. As a result, 1976 (286 up- and 1690 down-regulated), 887 (159 up- and 728 down-regulated), and 297 (22 up- and 72 down-regulated) DEGs were identified in RB vs. CK, UVB vs. CK, and UVB vs. RB, respectively (Figure 4B). Moreover, 1173, 226, and 99 unique DEGs were identified in RB vs. CK, UVB vs. CK, and UVB vs. RB, respectively, and 14 DEGs existed in all three comparison groups (Figure 4C).

Gene Ontology (GO) and Kyoto Encyclopedia of Genes and Genomes (KEGG) analyses were performed on the DEGs of strawberry fruits under different light treatments (Appendix A). The DEGs in RB vs. CK were divided into 45 function groups, including 21 biological process categories, 12 cellular component categories, and 12 molecular function categories. Moreover, the DEGs in UVB vs. CK were further classified into 36 functional groups, including 18 biological process categories, 9 cellular component categories, and 9 molecular function categories. The DEGs in UVB vs. RB were divided into 36 functional groups, including 15 biological process categories, 10 cellular component categories, and 11 molecular function categories. Among the biological processes, the top three GO terms were cell process, single-organism process, and metabolic process. Within the cell component category, cell part, cell, and membrane were the terms with the greatest abundance. Under the molecular function category, binding, catalytic activity, and transporter activity were the most highly represented terms. KEGG analysis of the DEGs in strawberry fruits under RB and UVB irradiation showed that biosynthesis of secondary metabolites (ko01110) and metabolic pathways (ko01100) was significantly enriched, indicating that both RB and UVB irradiation could result in dynamic changes of secondary metabolites and affect the fruit quality by regulating metabolic pathways and synthesizing metabolites.

### 2.4. Differential Expression of Flavonoid Biosynthetic and Regulatory Genes in Strawberry Fruits under RB and UVB Irradiation

In the present study, genes encoding key enzymes involved in flavonoid biosynthesis were identified based on KEGG analysis and gene function annotation (Figure 5). As a result, 37 genes (Appendix A), including 2 *PAL* genes, 1 *C4H* gene, 7 *4CL* genes, 1 *CHS* gene, 2 *CHI* genes, 1 *F3H* gene, 2 *FLS* genes, 3 *DFR* genes, 3 *ANS* genes, 1 *F3′H* gene, 1 *F3′5′H* gene, 1 *OMT* gene, and 12 *UFGT* genes, were identified in RB vs. CK and UVB vs. CK. Of these, 25 genes (1 *PAL* gene, 1 *C4H* gene, 6 *4CL* genes, 1 *CHS* gene, 2 *CHI* genes, 1 *F3H* gene, 3 *DFR* genes, 2 *ANS* genes, 1 *F3′H* gene, 1 *F3′5′H* gene, 1 *OMT* gene, and 5 *UFGT* genes) and 12 genes (1 *PAL* gene, 1 *4CL* gene, 1 *CHS* gene, 2 *CHI* genes, 2 *FLS* genes, 1 *ANS* gene, and 7 *UFGT* genes) were up-regulated and down-regulated in RB vs. CK, respectively. Moreover, 27 genes (2 *PAL* genes, 1 *C4H* gene, 6 *4CL* genes, 1 *CHS* gene, 2 *CHI* genes, 1 *F3H* gene, 1 *FLS* gene, 2 *DFR* genes, 1 *ANS* genes, 1 *F3′H* gene, 1 *F3′5′H* gene, 1 *OMT* gene, 7 *UFGT* genes) and 10 genes (1 *4CL* gene, 1 *FLS* gene, 1 *DFR* gene, 2 *ANS* genes, and 5 *UFGT* genes) were up-regulated and down-regulated in UVB vs. CK, respectively. Notably, five genes, including one *C4H* gene (*FvH4_3g40570*), three *4CL* genes (*FvH4_4g09340*, *FvH4_6g27940*, and *FvH4_2g37640*), and one *UFGT* gene (*FvH4_7g16110*), were both up-regulated in RB vs. CK and UVB vs. CK, suggesting that these genes are induced by RB and UVB light to promote the biosynthesis of flavonoids and anthocyanins.

It is well known that transcription factors (TFs) play essential roles in regulating the expression of most structure genes involved in the flavonoid biosynthesis pathway. In this study, a total of 60 TFs belonging to the 25 TF families with differential expressions were identified in this RNA-seq data (Appendix A and Figure 6A). Of these TFs, 55 TFs showed similar expression patterns to most structure genes of flavonoid biosynthesis, which were up-regulated in RB vs. CK, while 5 TFs—including 1 GRF, 2 NACs, 1 WRLY, and 1 RITF1—were down-regulated in RB vs. CK (Figure 6B). Moreover, 29 TFs were up-regulated in UVB vs. CK, while only 1 NAC (*FvH4_1g25150*) was down-regulated in UVB vs. CK (Figure 6C). Furthermore, our results showed that 28 TFs were up-regulated in RB vs. CK and UVB vs. CK, suggesting that these TFs played positive regulatory roles in response to RB lights and UVB (Figure 6D). One NAC was down-regulated in RB vs. CK and UVB vs. CK, suggesting that this TF played negative regulatory roles in the response to RB lights and UVB.

### 2.5. Correlation Analysis between Transcripts and Flavonoid Derivatives

To further explore the regulatory network of flavonoid and anthocyanin biosynthesis in strawberry fruits, correlation tests between quantitative changes of flavonoids and anthocyanins and transcripts were performed in strawberry fruits under RB and UVB irradiation. Based on the transcripts and DEMs in RB vs. CK enriched in the flavonoid biosynthesis pathway, our results showed that 23 structure genes in the flavonoid biosynthesis pathway displayed higher correlation coefficient values (*r* > 0.85), with 20 flavonoids and their interaction networks shown in Figure 7A. The 23 structure genes included 1 *PAL* gene, 4 *4CL* genes, 2 *ANS* genes, 1 *C4H* gene, 2 *CHI* genes, 1 *CHS* gene, 3 *DFR* genes, 1 *FLS* gene, 1 *OMT* gene, and 7 *UFGT* genes. The 20 flavonoids contained 1 chalcones, 4 flavanones, 5 flavones, and 10 flavanols. Notably, one *4CL* (*FvH4_5g10440*) was positively correlated with four flavonols, including Kaempferol-3-O-galactoside-4′-O-glucoside, Epicatechin-3′-O-β-D-glucopyranoside, catechin-4-β-D-galactopyranoside, and Kaempferol-3-O-(4″-p-coumaroyl)rhamnoside. The DFR gene (*FvH4_1g13990*) displayed high correlation with four flavonoids (Kaempferol-3-O-(4″-p-coumaroyl)rhamnoside, Luteolin-7-O-neohesperidoside (Lonicerin), (Isorhamnetin-3-O-rutinoside(Narcissin), and Myricetin-3-O-rhamnoside). The *UFGT* gene (*FvH4_7g20870*) was significantly correlated with three flavonoids, Luteolin-7-O-glucoside, Luteolin-8-C-arabinoside, and Kaempferol-3-O-(4″-p-coumaroyl)rhamnoside. Furthermore, the correlation analysis between the differentially expressed TFs and DAMs was also conducted in this study. We found 27 TFs with a high correlation with 19 flavonoids (*r* > 0.85, Figure 7B). The 27 TFs included 1 AP2, 1 AUX/IAA, 1 bZIP, 4 Dofs, 1 GARS, 3 HD-ZIPs, 1 HMG, 1 HSF, 5 MYBs, 4 NACs, 1 NF-YA, 1 NF-YB, 1 Trihelix, 2 WRKYs, and 1 HPK. The 19 DAMs contained 1 chalcones, 4 flavanones, 5 flavones, and 9 flavanols. Among them, a bZIP TF, *FvH4_2g36400*, displayed high correlation with 6 flavonoids, Myricetin-3-O-rhamnoside, Luteolin-7-O-glucoside, Kaempferol-3-O-(4″-p-coumaroyl)rhamnoside, catechin-4-β-D-galactopyranoside, Epicatechin-3′-O-β-D-glucopyranoside, and Isorhamnetin-3-O-rutinoside (Narcissin).

Based on the transcripts and DEMs in UVB vs. CK enriched in the flavonoid biosynthesis pathways, 10 structure genes displayed a high correlation with 31 flavonoids, and their interaction networks are shown in Figure 7A. The 10 structure genes included 1 *PAL* gene, 2 *4CL* genes, 1 *ANS* gene, 1 *C4H* gene, 1 *F3H* gene, 1 *FLS* gene, and 3 *UFGT* genes. The 30 flavonoids contained 1 chalcone, 1 anthocyanin, 2 flavanols, 4 flavanones, 7 flavones, and 16 flavanols. Notably, the *4CL* gene (*FvH4_4g09340*) and *C4H* (*FvH4_3g40570*) displayed a higher correlation with 28 flavonoids (Appendix A). Moreover, correlation analysis between TFs and DAMs was also performed, showing that the 10 TFs displayed a higher correlation with 23 flavonoids. The ten TFs included one AP2, one AUX/IAA, one Dof, one HD-ZIP, one HMG, one MYB, two NACs, and one WRKY. The 23 flavonoids contained 1 anthocyanin, 1 chalcone, 1 flavanol, 7 flavanones, 5 flavones, and 8 flavones. The AP2 (*FvH4_1g21210*) and AUX/IAA (*FvH4_2g22520*) displayed a higher correlation with 8 and 14 flavonoids (Appendix A).

## 3. Materials and Methods

### 3.1. Plant Materials and Treatments

The cultivated strawberry (*F. × ananassa* “*Benihoppe*”) was used in this study. Two-month-old strawberry seedlings (about 20 cm in height) were cultivated in pots with a volume of 4.5 L, which contained a substrate that consisted of a mixture of vermiculite, vermicompost, and soil (2:2:6). The potted strawberries, 7 days after flowering (7 DAF), were selected and divided into three groups. The selected strawberries were transferred to a greenhouse with controlled environmental conditions (16 h photoperiod with a light intensity of 100 μmol m^−2^ s^−1^ at 25 °C, 8 h dark at 16 °C, and 75% relative humidity). Experiment Ⅰ: Strawberry plants at 7 DAF were selected and irradiated with red light supplemented with blue light (RB; red/blue light photosynthetic photon flux density (PPFD) ratio was 7/1) at 300 mol m^−2^ s^−1^ PPFD for 7 weeks at 25 °C. Experiment II: Strawberry plants at 7 DAF were selected and irradiated with UVB light for 7 weeks at 25 °C. Strawberry plants at 7 DAF irradiated with natural light were used as the control group (CK). The RB and UVB lights (30 W, Hangzhou Small Sun Agricultural Science and Technology Co., Ltd., Hangzhou City, China) installed at the top of the chambers were applied to treat strawberry seedlings until samples were collected. Strawberry fruits were sampled at 30 (white stage), 37 (pink stage), and 45 (red stage) days after flowering. The collected samples were immediately frozen in liquid nitrogen. The collected strawberry fruits were stored at −80 °C for further analysis. Each treatment had 3 biological replicates, and each replicate had 15 fruits from the same stage.

### 3.2. Determination of the Content of Flavonoids and Anthocyanins

The total flavonoid contents of the crude extract were measured using the aluminum chloride colorimetric method. A total of 50 µL of ethanol extract of strawberry fruit (1 mg/mL ethanol) was added to 1 mL with methanol. Then, 4 mL of distilled water and 0.3 mL of 5% NaNO_2_ solution were added to the above mixture. After 5 min of incubation, 0.3 mL of 10% AlCl_3_ solution was added and mixed. The mixture was left to stand for 6 min at room temperature, and then 2 mL of 1 mol/L NaOH solution was added. Double-distilled water was used to increase the volume up to 10 mL of mixture and stand for 15 min. The absorbance of this solution was determined at 510 nm.

Anthocyanins were extracted from strawberry fruits using the procedures that have been described previously [50]. Quantification for total anthocyanins was carried out according to the pH differential method using a UV–visible spectrophotometer (Shimadzu, Kyoto, Japan) at 496 nm and 700 nm [51]. The total monomeric anthocyanin concentration was calculated as pelargonidin 3-glucoside, which was demonstrated to be the major anthocyanin in strawberries [52].

### 3.3. Determination of Titratable Acidity, Total Sugar Content, and Lignin Content

The content of titratable acidity (TA) of strawberry fruits was determined by means of the titration method explained by Teka (2013) [53]. About 5 mL of strawberry fruit juice was taken and diluted with 95 mL of distilled water and then titrated to pH 8.2 using 0.1 N of NaOH. The total sugar contents were measured according to the method by Liao et al. (2019) [54]. The strawberry fruits at three developmental stages were dried at 105 °C for 1 h and then at 65 °C for 24 h. The fruit samples were ground into a fine powder. The lignin contents were determined based on the method previously described by Ma et al. (2021) [55].

### 3.4. Antioxidant Enzyme Assays

Strawberry fruit samples (0.1 g) were ground into a fine powder in liquid nitrogen and then homogenized with 10 mL of extraction buffer. The extraction buffer contained 0.1% Triton-X-100, 1 mM of ethylene diamine tetraacetic acid (EDTA), 100 mM of phosphate buffer (pH 7.0), and 1% polyvinyl pyrrolidone (PVP). The homogenate was spun for 20 min at 12,000 rpm at 4 °C. The supernatant was used to determine the antioxidant enzyme activities.

The activities of SOD, POD, and CAT enzymes were determined according to the method of Yan et al. (2023), with some modifications [56]. An enzyme extract (0.2 mL) was added to a 3 mL reaction mixture (50 mM of phosphate buffer, 75 mM of nitroblue tetrazolium, 13 mM of methionine, 0.1 mM of EDTA, and 4 mM of riboflavin) and placed in a growth chamber with the light intensity of 100 μmol m^−2^ s^−1^ for 20 min. Controls were performed in the same manner but with a light-proof treatment. The absorbance of the reaction mixture was measured at 560 nm for all samples. One unit of SOD activity (U) was defined as the amount of enzyme that produced 50% inhibition of nitroblue tetrazolium reduction under assay conditions.

Enzyme extracts (20 µL) were added to a 180 µL reaction mixture (50 mM of phosphate buffer (pH 7.0), 10 mM of guaiacol, and 10 mM of hydrogen peroxide). The POD activity was assayed by measuring the change of absorption at 470 nm due to guaiacol oxidation. One unit of POD activity was defined as the absorbance change of 0.01 per minute at 470 nm.

CAT activity was measured by following the consumption of H_2_O_2_ at 240 nm for 4 min. One unit of CAT activity was defined as the absorbance change of 0.01 per minute at 240 nm.

### 3.5. Determination of the Malonaldehyde (MDA) Content

The MDA content was determined based on the method of Wang et al. (2023), with several modifications [57]. The strawberry fruits (100 mg) were crushed into powder with liquid nitrogen and then dissolved thoroughly with 10 mL of extraction buffer (100 mM of pH 7.0 phosphate buffer, 0.1% TritonX-100, 1 mM of EDTA, and 1% PVP). The mixture was spun for 20 min at 12,000 rpm at 4 °C. Then, 1.8 mL of 0.5% (*w*/*v*) thiobarbituric acid was added to 0.2 mL of supernatant and boiled for 20 min. The mixture was cooled at room temperature and centrifuged at 3000 rpm for 5 min. The supernatant was measured at 532 nm, 600 nm, and 450 nm using a microplate reader. MDA content (μmol/g) = [6.45 × (A532₋A600) − 0.56 × A450]/fresh weight.

### 3.6. RNA Extraction, Library Construction, RNA Sequencing, and Data Analysis

The strawberry fruits treated with different LED lights were used to conduct RNA-seq. The total RNA of strawberry fruits was extracted using TaKaRa MiniBEST Plant RNA Extraction Kit (TaKaRa, Osaka, Japan). The RNA quality and concentration were determined using a Bioanalyzer 2100 system with an RNA 6000 Nano kit (Agilent, Santa Clara, CA, USA). Isolation of mRNA was performed using the NEBNext^®^ Poly(A) mRNA Magnetic Isolation Module (E7490). Construction of the cDNA library was performed based on the manufacturer’s instructions for NEBNext Ultra RNA Library Prep Kit for Illumina (NEB, E7530, San Diego, CA, USA) and NEBNext Multiplex Oligos for Illumina (NEB, E7500). The constructed cDNA libraries of strawberry fruits were sequenced on an Illumina HiSeq™ sequencing platform, and the clean reads were mapped to the reference genome of *F. × ananassa* using HISAT v2.1.0.

Differentially expressed genes (DEGs) between two groups were analyzed using DESeq2, and genes with a fold change (FC) of ≥2 and a false discovery rate of <0.01 were considered as DEGs. Seven databases—Nr (NCBI non-redundant protein sequences), Nt (NCBI non-redundant nucleotide sequences), GO (Gene Ontology), KO (Kyoto Encyclopedia of Genes and Genomes Ortholog database), KOG/COG (Clusters of Orthologous Groups of proteins), Pfam (Protein family), and Swiss-Prot (a manually annotated and reviewed protein sequence database)—were used for gene function annotation. The transcripts’ raw data were deposited in the NCBI SRA database under the project number.

### 3.7. Metabolite Profiling

The fruit samples (100 mg) were ground into powder using liquid nitrogen, and the homogenate mix was resuspended with prechilled 80% methanol using a well vortex. The mix was chilled on ice for 5 min and then centrifuged (15,000× *g*, 4 °C, and 20 min). Some of the supernatant was transferred to a new EP tube and diluted to a final concentration containing 53% methanol by adding LC-MS-grade water. Then, the mix was centrifuged at 15,000× *g*, 4 °C for 20 min, and injected into the LC-MS/MS system analysis.

UHPLC-MS/MS analyses were conducted using a Vanquish UHPLC system (Thermo Fisher, Waltham, MA, USA) connected to an Orbitrap Q ExactiveTM HF mass spectrometer (Thermo Fisher, Waltham, MA, USA) in Novogene Co., Ltd. (Beijing, China). Samples were injected onto a Hypesil Gold column (100 × 2.1 mm, 1.9 μm) using a 17 min linear gradient at a flow rate of 0.2 mL/min. The eluents for the positive polarity mode were eluent A (0.1% FA in Water) and eluent B (Methanol). The eluents for the negative polarity mode were eluent A (5 mM ammonium acetate, pH 9.0) and eluent B (methanol). The solvent gradient was set as follows: 2% B, 1.5 min; 2–100% B, 3 min; 100% B, 10 min; 100–2% B, 10.1 min; 2% B, 12 min. Q ExactiveTM HF mass spectrometer was operated in positive/negative polarity mode with a spray voltage of 3.5 kV, capillary temperature of 320 °C, sheath gas flow rate of 35 psi, aux gas flow rate of 10 L/min, S-lens RF level of 60, and aux gas heater temperature of 350 °C.

### 3.8. Statistical Analysis

The data were analyzed by means of one-way analysis of variance (ANOVA) and Duncan’s multiple range test (*p* < 0.05) using SPSS 25.0 statistics (SPSS Inc., Chicago, IL, USA). The cluster heatmap analysis was conducted using software available online at https://cloud.metware.cn/toolCustom/3 (accessed on 3 April 2024). Principal component analysis (PCA) and orthogonal partial least squares discriminant analysis (OPLS-DA) were carried out using online software (https://www.omicshare.com/tools/Home/Soft/getsoft, accessed on 3 April 2024). Correlation analysis of 9 samples was performed using TBTOOLS (TBTOOLS version no. 2.007: An Integrative Toolkit Developed for Interactive Analyses of Big Biological Data). Graphs were drawn using OriginPro 2021 (OriginLab, Northampton, MA, USA).

## 4. Discussion

Flavonoids are a group of natural polyphenol substances that are abundant in vegetables and fruits and are responsible for their color, fragrance, and flavor characteristics [58]. As secondary plant metabolites, flavonoids play essential roles in many biological processes and responses to environmental factors in plants [59,60]. Numerous studies have revealed that flavonoid biosynthesis is not only regulated by internal genetic factors but also affected by external factors such as light, temperature, and nutritional levels [59,61]. Among them, light is one of the most important environmental factors affecting flavonoid biosynthesis in plants, which promotes fruit ripening and coloration [62]. Our results showed that RB and UVB light promoted fruit coloration, increased the contents of flavonoids and anthocyanins, and enhanced the activities of antioxidant enzymes. These results were consistent with previous studies [35,36,63]. Gam et al. (2020) revealed that BR (one blue, four red) promoted the growth and flavonoid accumulation of *Anoectochilus roxburghii* by enhancing the expression of several related genes [63]. Zhang et al. (2018) found that BR (one blue, one red) promoted earlier strawberry fruit coloration by increasing the total anthocyanin content [35,36]. In addition, previous studies have proven that UVB induces the expression of genes involved in the flavonoid biosynthetic pathway, thus increasing the anthocyanin content of various plants, such as apples [19], blueberries [20], grapes [21], and carrot taproots [22]. Moreover, our results showed for the first time that the production of flavonoids was more pronounced in fruits irradiated with UVB than in fruits irradiated with RB light, suggesting that flavonoids might be used as protective substances against UVB radiation.

In the present study, metabolomics analysis showed that 16 phenolic acids and 17 flavonoids were decreased in RB vs. CK. The 25 phenolic acids and 36 flavonoids were decreased in UVB vs. CK. These results further indicated that RB and UVB light enhanced antioxidant activity and improved fruit quality by increasing contents of phenolic acids and flavonoids, respectively. Previous studies have revealed that combining red and blue LED light improves growth and phenolic acid contents in many plants, such as *Salvia miltiorrhiza*, *Melissa officinalis*, and *Nasturtium officinale* [64,65,66]. UVB treatment accumulated phenolic acids in barley seedlings. Zhang et al. (2023) revealed that UVB enhanced the phenolic and flavonoid contents and antioxidant activity of green- and red-leaf lettuce cultivars [67]. In addition, the number of phenolic acids and flavonoids down-regulated in UVB vs. CK is significantly higher than in RB vs. CK, which has been revealed for the first time in our research. This result indicated that phenolic acids and flavonoids effectively suppressed UVB-induced elevation of intracellular ROS to protect the plant against oxidative damage. In addition, two and six anthocyanins were significantly enriched in strawberry fruits under RB and UVB irradiation, consistent with the previous studies showing that RB and UVB can induce the accumulation of anthocyanins.

As is well known, flavonoid biosynthesis is regulated by both structural genes and TFs. In our study, 37 genes encoding key enzymes involved in flavonoid biosynthesis (e.g., *PAL*, *C4H*, *4CL*, *CHS*, *CHI*, *F3H*, *FLS*, *DFR*, *ANS*, *F3′H*, *F3′5′H*, *OMT*, and *UFGT* genes) were identified by means of KEGG analysis and gene function annotation. This result is consistent with previous studies and further validates the metabolite analysis results of metabolomics analysis that phenolic acids and flavonoids significantly accumulate in UVB-treated fruits [68,69,70,71,72]. Interestingly, previous metabolite analysis revealed that RB and UVB induced the production of anthocyanins and promoted fruit coloration [73,74,75]. It has been known that flavonoid-3-*O*-glucosyltransferase genes played essential roles in anthocyanin biosynthesis, which is essential for the formation of colored anthocyaninsin. In the present study, 12 flavonoid-3-*O*-glucosyltransferase genes were differentially expressed in strawberry fruits treated with RB and UVB. Seven *UFGT* and five *UFGT* genes were down-regulated in RB vs. CK and UVB vs. CK, respectively. This result suggests that flavonoid-3-*O*-glucosyltransferase genes might catalyze the generation of flavonoids and anthocyanins in the glycosylated form. In addition, the TFs played essential roles in regulating flavonoid and anthocyanin biosynthesis. In this study, 55 and 29 TFs displayed similar expression patterns to structure genes in flavonoid biosynthesis pathways that were up-regulated in RB vs. CK and UVB vs. CK, respectively. Five and one TFs displayed expression patterns contrary to most structure genes of flavonoid biosynthesis, which were down-regulated in RB vs. CK and UVB vs. CK, respectively. These results indicated that these genes played positive or negative regulatory roles in regulating flavonoid and anthocyanin biosynthesis.

Correlation analysis between metabolome and transcriptome analysis revealed that the accumulation of flavonoids and anthocyanin was closely related to the structure genes and TFs, indicating that RB and UVB promoted the accumulation of flavonoids by regulating the expression of corresponding structural genes and TFs. Notably, there was one member of bZIP TFs (*FvH4_2g36400*) in RB vs. CK, a member of clade C of the bZIP family, which was closely related to six flavonoids (Myricetin-3-O-rhamnoside, Kaempferol-3-O-(4″-p-coumaroyl)rhamnoside, catechin-4-β-D-galactopyranoside, Isorhamnetin-3-O-rutinoside, Luteolin-7-O-glucoside, and Epicatechin-3′-O-β-D-glucopyranoside. A member of clade C of the grapevine bZIP family, *VvibZIPC22*, was able to activate the transcriptional expression of specific genes of the flavonoid pathway, including *VviCHS3*, *VviCHI*, *VviFLS1*, and *VviANR.* Based on the previous transcriptome analysis, we found that the expression pattern of *FvH4_2g36400* was consistent with the expression of 25 genes (one *PAL*, one *C4H*, six *4CLs*, one *CHS*, two *CHI*, one *F3H*, three *DFRs*, two *ANSs*, one *F3′H*, one *F3′5′H*, one *OMT*, and five *UFGTs*). Moreover, the *AP2* (*FvH4_1g21210*) in UVB vs. CK, the homologous gene of *Arabidopsis AtERF4*, was found to be closely related to eight flavonoids, Phloretin-4′-O-(6″-p-Coumaroyl)glucoside, Eriodictyol-8-C-glucoside-4′-O-glucoside Apigenin-7-O-Gentiobioside,Quercetin-3-O-(2″′-O-p-coumaroyl)sophoroside-7-O-glucoside, Vitexin-2″-O-galactoside, Tricin-4′-O-syringic acid, Vitexin-2″-O-rhamnoside, and C-glucosyl-C-arabinosyl-2-hydroxynaringenin. In plants, ERF4 promotes the biosynthesis of chlorogenic acid and flavonoids. In the previous transcriptome analysis, we found that the expression pattern of *FvH4_1g21210* was consistent with the expression of 27 genes (2 *PALs*, 1 *C4H*, 6 *4CLs*, 1 *CHS*, 2 *CHIs*, 1 *F3H*, 1 *FLS*, 2 *DFRs*, 1 *ANS*, 1 *F3′H*, 1 *F3′5′H*, 1 *OMT*, and 7 *UFGTs*). Thus, these results suggested that bZIP (*FvH4_2g36400*) and AP2 (*FvH4_1g21210*) are induced by RB and UVB, respectively, thereby promoting the biosynthesis of flavonoids and anthocyanins.

## 5. Conclusions

In the present study, RB and UVB light promoted fruit coloration and the accumulation of flavonoids and anthocyanins in strawberry fruits, which also increased the activities of antioxidant enzymes. Metabolomics analysis showed that phenolic acids, flavonoids, and anthocyanins were differentially accumulated in strawberry fruits irradiated with RB and UVB light. The essential structure genes and TFs involved in regulating the differential accumulation of flavonoids and anthocyanins in strawberry fruits treated with RB and UVB light were identified based on RNA-seq data. Furthermore, the correlation analysis between RNA-seq data and metabolite profiling further revealed that two TFs (*bZIP*, *FvH4_2g36400*; *AP2*, *FvH4_1g21210*) induced by RB and UVB irradiation, respectively, exhibited a similar expression pattern to most structural genes, which was closely correlated with six and eight flavonoids, respectively. These results deepen our understanding of the composition and corresponding molecular mechanisms of flavonoid and anthocyanin biosynthesis in strawberry fruits under RB and UVB light. Furthermore, the regulatory mechanisms of RB and UVB involved in the biosynthesis of flavonoids and anthocyanins in strawberry fruits differ.

## Figures and Tables

**Figure 1 plants-13-01070-f001:**
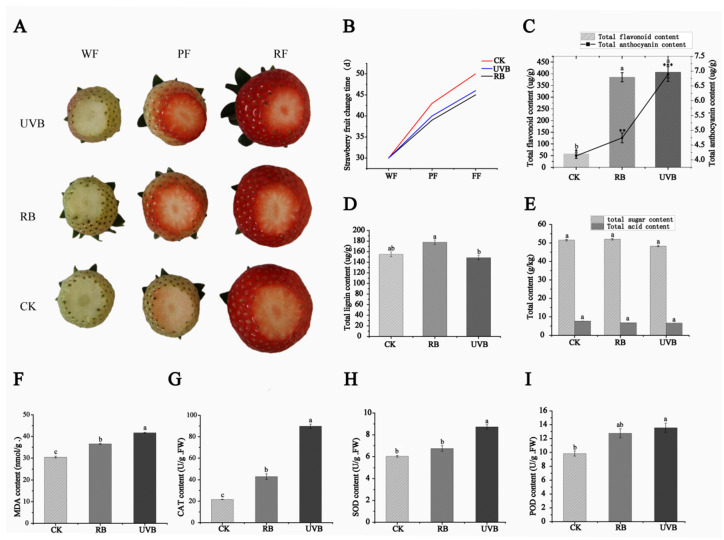
Morphological traits and physiological indicators under RB and UVB light. (**A**) Morphological traits in strawberry fruits irradiated with RB and UVB light. (**B**) Effects of RB light on strawberry fruit coloration. Fruits were harvested and photographed in three developmental stages. WF, white fruit; PF, partial red fruit; RF, full red fruit. (**C**) The contents of flavonoids and anthocyanins (**, *** significantly increased in strawberry fruits under RB and UVB light treatments compared to control group (* CK)). (**D**) The contents of lignins. (**E**) The contents of titratable acidity and total sugar. (**F**) The contents of malondialdehyde (MDA). (**G**–**I**) Activities of superoxide dismutase (SOD), catalase (CAT), and peroxidase (POD). The values are means of three replicates. CK, UVB, and RB represent strawberry fruits irradiated with natural light, ultraviolet B light, and red light supplemented with blue light. Vertical bars indicate the mean standard error. Lowercase letters indicate significant differences between the control group and experimental group at *p* ≤ 0.05 according to Duncan’s multiple range test.

**Figure 2 plants-13-01070-f002:**
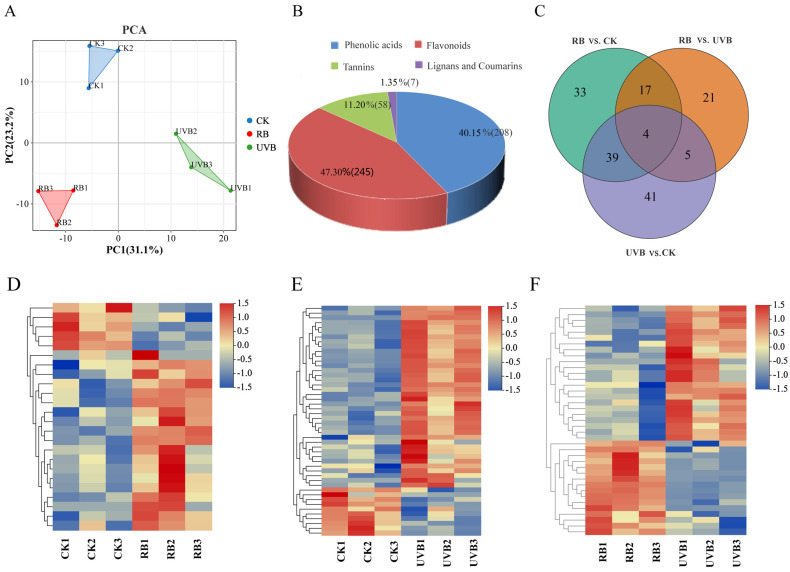
Principal component analysis (PCA) and differential accumulation of metabolites (DAMs) in strawberry fruit under RB and UVB light. (**A**) Principal component analysis (PCA) of metabolites detected in strawberry fruits; (**B**) The classification of DAMs; (**C**) Venn diagram of DAMs in different comparison groups; panels (**D**–**F**) represent the heat maps of differential accumulation of flavonoid metabolites RB vs. CK, UVB vs. CK, and RB vs. CK comparison groups, respectively. RB, UVB, and CK represent red/blue LED light treatment, UVB treatments, and nature light treatment, respectively.

**Figure 3 plants-13-01070-f003:**
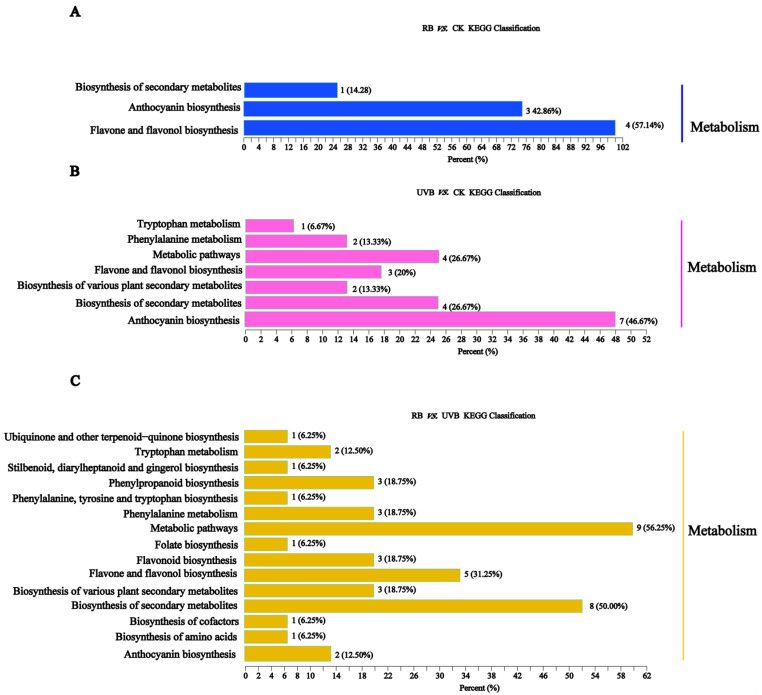
KEGG classification of differential flavonoids. (**A**) RB vs. CK; (**B**) UVB vs. CK; (**C**) RB vs. UVB.

**Figure 4 plants-13-01070-f004:**
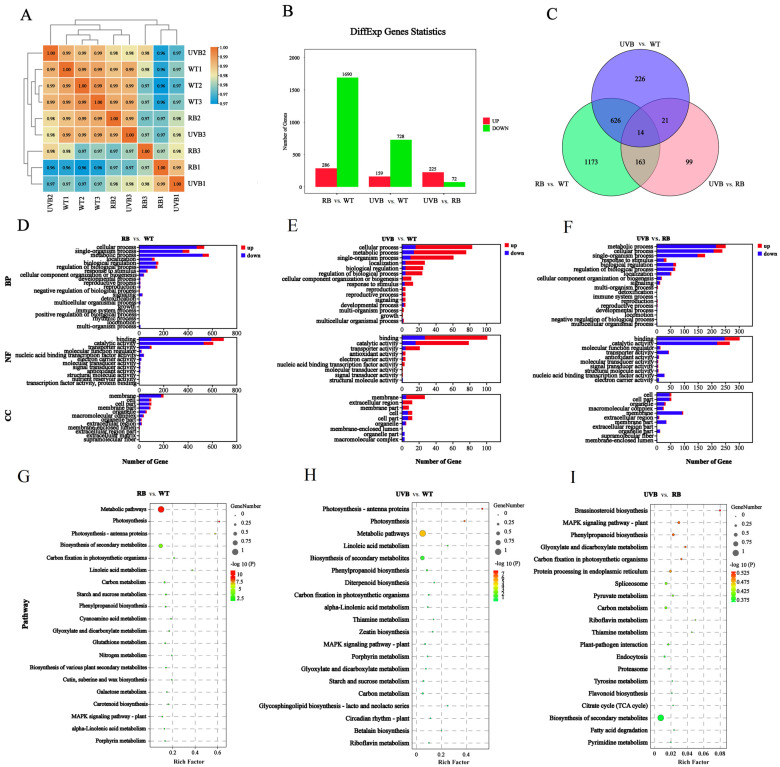
Number, GO enrichment, and KEGG pathway enrichment of DEGs identified in strawberry fruits treated with RB and UVB irradiation. (**A**) Sample correlation analysis. (**B**) The number of DEGs in different comparison groups. (**C**) Venn diagram of differential metabolites in different comparison groups. (**D**–**F**) Gene ontology enrichment analysis of the DEGs in RB vs. CK (**D**), UVB vs. CK (**E**), and RB vs. UVB (**F**). (**G**–**I**) KEGG enrichment analysis of the DEGs in RB vs. CK (**G**), UVB vs. CK (**H**), and RB vs. UVB (**I**).

**Figure 5 plants-13-01070-f005:**
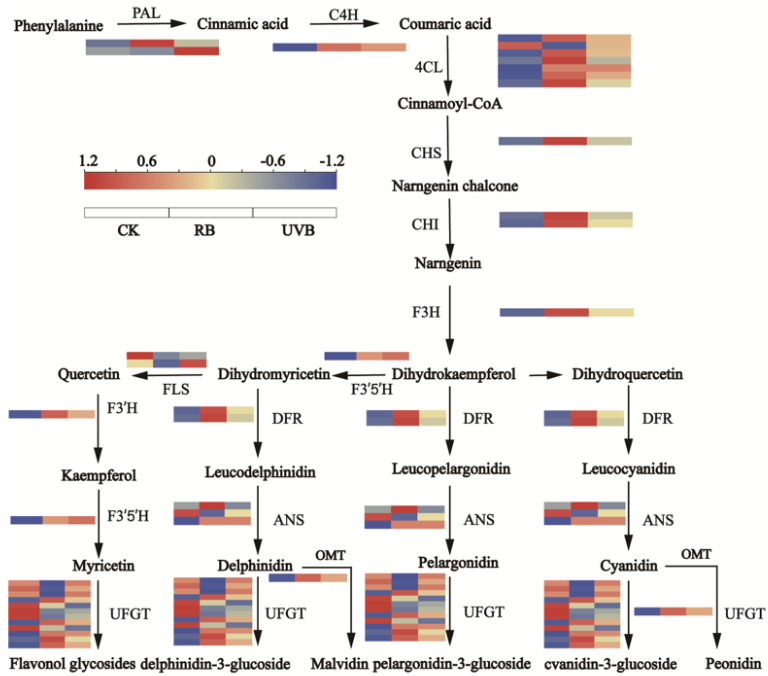
The biosynthetic pathway of flavonoids and the heat map of changes in transcripts in flavonoid metabolism. Rectangles marked with red and yellow backgrounds represent increased abundances of transcripts in RB vs. CK and UVB vs. CK, respectively. The blue background represents reduced abundances of transcripts in RB vs. CK and UVB vs. CK.

**Figure 6 plants-13-01070-f006:**
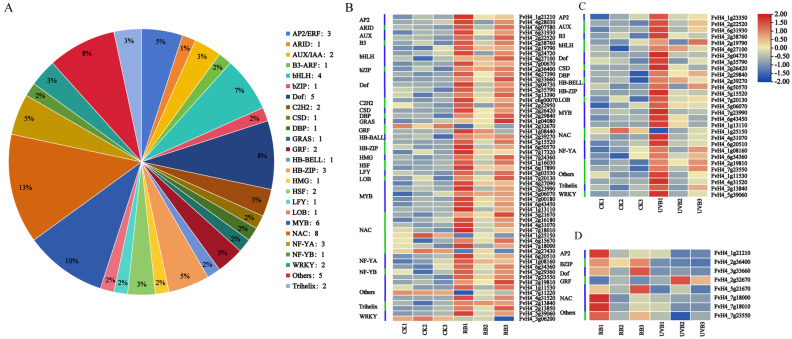
The differentially expressed TFs in strawberry fruits. (**A**) The number of differentially expressed TFs of different types. (**B**) The differentially expressed TFs in RB vs. CK. (**C**) The differentially expressed TFs in UVB vs. CK. (**D**) The differentially expressed TFs in RB vs. UVB. A heat map of dynamic changes of differentially expressed TFs associated with flavonoid biosynthesis. Rectangles marked with red and blue backgrounds represent increased and reduced abundances of TF transcripts, respectively.

**Figure 7 plants-13-01070-f007:**
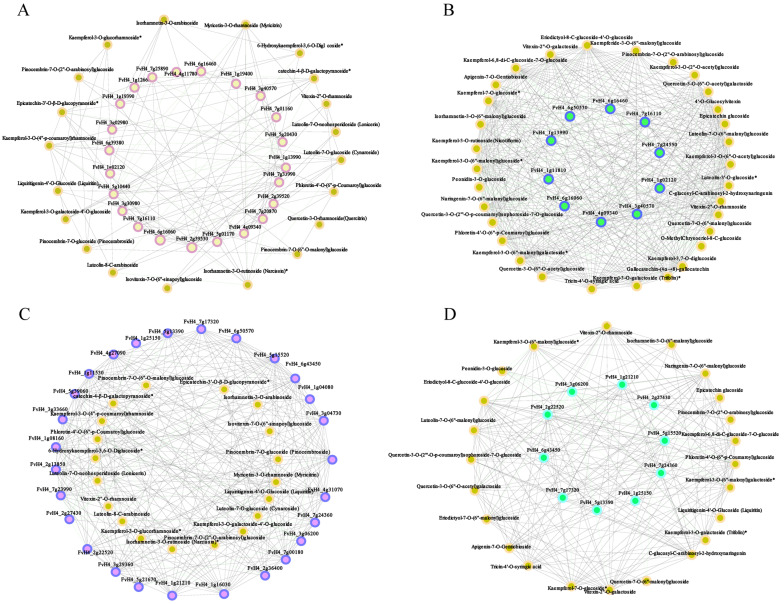
The connection network analysis between flavonoid-related genes and differential accumulation of flavonoid metabolites in strawberry fruits under RB and UVB irradiation. (**A**) The connection network between structural genes associated with flavonoid biosynthesis and differential flavonoids in RB vs. CK. Yellow and bright blue represent flavonoids and structural genes, respectively. (**B**) The connection network between structural genes associated with flavonoid biosynthesis and differential flavonoids in UVB vs. CK. Yellow and bright blue represent flavonoids and structural genes, respectively (**C**) The connection network between differentially expressed TFs and differential flavonoids in RB vs. CK. The yellow circle with purple edges and the yellow circle represent differentially expressed TFs and differential flavonoids, respectively. (**D**) The connection network between differentially expressed TFs and differential flavonoids in UVB vs. CK. The yellow circle with purple edges and the purple circle with blue edges represent differentially expressed TFs and differential flavonoids, respectively.

## Data Availability

Data are contained within the article and Appendix A.

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
