# Peer review of "Effects of Supplemental Lighting on Flavonoid and Anthocyanin Biosynthesis in Strawberry Flesh Revealed via Metabolome and Transcriptome Co-Analysis"

_plants, 2024, doi:10.3390/plants13081070_

Round 1

Reviewer 1 Report

Comments and Suggestions for Authors

The results presented in the manuscript are very valuable and interesting. However, the work in its current form cannot be published and requires corrections, including linguistic corrections. The authors should carefully read the entire text of the manuscript again - there are many spelling errors. I list the necessary corrections and suggestions below:

Abstract:

1)     Line 1: „many plants” („s” is missing) 

2)     Line 2: should be: “the detailed composition of flavonoids” 

3)     Line 5: sentence shouldn’t start with “and” 

4) This fragment should be rewritten: “Of these, 18 phenolic acids, 23 flavonoids, and 8 anthocyanins were differ-entially accumulated in the strawberry fruits irradiated with red/blue (RB). And, 30 phenolic acids, 46 flavonoids, and 9 anthocyanins were differentially accumulated in the strawberry fruits irradi-ated with UVB.” – it is not clear. What does it mean different? Different means how? Maybe you should make these fragment shorter like: 

“18 phenolic acids, 23 flavonoids, and 8 anthocyanins were differentially accumulated in the strawberry fruits irradiated with red/blue (RB) comparing to other 30 phenolic acids, 46 flavonoids, and 9 anthocyanins fruits irradiated with UVB”. 

5)     In the abstract you should write smth about methodology before you write about results. How many light conditions did you test? What kind of light it was? LED light? You didn’t write anything about that.

Please rewrite the abstract.

Introduction:

6)      Should be: (Fragaria ×ananassa) (sign × stick to the “ananassa” word). 

7)     According to my knowledge names of the botanical families we do not write in italics (Rosaceae). 

8)     Is the form of citations correct according to journal requirements? -> Arabic numbers in brackets stick to the last word in the sentence (with no space)? 

9) It should be: “Anthocyanin biosynthesis is also affected by (…)” 

10) I think thise fragment should be used in Discussion section: 

“For example, Liu et al. (2018) re-vealed that light quality affects flavonoid production by regulating the related gene ex-pression in Cyclocarya paliurus(15). Blue light could induces biosynthesis of flavonoids in Epimedium sagittatum(16). Kadomura-Ishikawa et al. (2013) has revealed that blue and red light quickly promoted the anthocyanin biosynthesis in fruits by regulating the expression of LAR and ANR genes(17). Illumination with blue light-emitting diodes (LEDs) induced the anthocyanins accumulation in red lettuce(18). Zhang et al.(2018a) has shown that blue and red light increased the contents of total anthocyanins, pelargonidin 3-glucoside, and pelargonidin 3-malonyl glucoside in strawberry fruits(19). Moreover, UV-B can promote flavonoid biosynthesis, leading to improved fruit quality of blueberry(3). Wang et al. (2009) found that the UV radiation increased the anthocyanin contents in blueberry(2).”

 11) “The objective of this study was to clarify the effects of red and blue light on strawberry fruit development and maturation.” – how about the UV? You write in the abstract about UV spectrum as well.

Materials and methods:

12) ExperimentI – space before I 

13) Please provide more technical information about LED lighting. What was the spectrum of the used lights in nanometers? What was PPFD? What kind of LED panels they were? How long during the day the supplementary lighting took place? In what time of the year the experiment took place? What other conditions were in the greenhouse (e.g. humidity). 

14)  Correct to: (p ≤ 0.05)  

Results and Discussion:

15) There is no full legend in Figure 1 (you should explain again the abbreviations such as CK, UVB, RB and WF, PF, RF – on chart B shouldn’t be RF instead of FF? And also it should be: ≤ sign in description under Figure. 

16) Spelling error: strawnerry (Line 4 of 3.1. chapter).

17) Sentence need to be rewritten: “The content of flavonoids and anthocyanins were significantly increased in strawnerry fruits promote the biosynthesis of flavonoids and anthocyanins in strawberry fruits under RB- and UVB light (Figure 1C).”

For example: The content of flavonoids and anthocyanins were significantly increased in strawberry fruits under RB- and UVB light (Figure 1C). 

19) Table S1), „ – delete space before coma 

20) Numerious  - spelling (Line 4 in Discussion section).

Comments on the Quality of English Language

 Extensive editing of English language required.

Author Response

Dear Reviewer,

On behalf of my co-authors, we thank you very much for giving us an opportunity to revise our manuscript. We have improved our manuscript based on the comments you provided. We appreciate for your warm work earnestly, and hope that the corrections will meet with approval. The responds to the comments are as follows:

Reviewer1

Comments and Suggestions for Authors

The results presented in the manuscript are very valuable and interesting. However, the work in its current form cannot be published and requires corrections, including linguistic corrections. The authors should carefully read the entire text of the manuscript again - there are many spelling errors. I list the necessary corrections and suggestions below:

Abstract:

1)Line 1: „many plants” („s” is missing)

Response: Thank you for your comments. We are sorry for the misunderstanding caused by our ignorance, in the revised manuscript, we have changed ‘plant’ to ‘plants’, please see Line 11:

Line 11: Light spectral composition influences the biosynthesis of flavonoids in many plants.

2) Line 2: should be: “the detailed composition of flavonoids”

Response: Thank you for your comments. We are sorry for the misunderstanding caused by our ignorance, in the revised manuscript, we have changed ‘the detailed the composition of flavonoids’ to ‘the detailed composition of flavonoids’, please see Lines 12-13:

Lines 12-13: However, the detailed composition of flavonoids and anthocyanins and their molecular basis for biosynthesis in strawberry fruits remain still unclear.

  • Line 5: sentence shouldn’t start with “and”

Response: Thank you for your comments. We are sorry for the misunderstanding caused by our ignorance, in the revised manuscript, we have revised this sentence, please see Lines 14-17:

Lines 14-17: Among them, 18 phenolic acids, 23 flavonoids, and 8 anthocyanins were differentially accumulated in the strawberry fruits irradiated with red/blue (RB) comparing to other 30 phenolic acids, 46 flavonoids, and 9 anthocyanins fruits irradiated with UVB. 30 phenolic acids, 46 flavonoids, and 9 anthocyanins were differentially accumulated in the strawberry fruits irradiated with UVB.

4) This fragment should be rewritten: “Of these, 18 phenolic acids, 23 flavonoids, and 8 anthocyanins were differ-entially accumulated in the strawberry fruits irradiated with red/blue (RB). And, 30 phenolic acids, 46 flavonoids, and 9 anthocyanins were differentially accumulated in the strawberry fruits irradi-ated with UVB.” – it is not clear. What does it mean different? Different means how? Maybe you should make these fragment shorter like:

Response: Thank you for your comments. We are sorry for the misunderstanding caused by our ignorance, in the revised manuscript, we have revised this sentence, please see Lines 14-17:

Lines 14-17: Among them, 18 phenolic acids, 23 flavonoids, and 8 anthocyanins were differentially accumulated in the strawberry fruits irradiated with red/blue (RB) comparing to other 30 phenolic acids, 46 flavonoids, and 9 anthocyanins fruits irradiated with UVB. 30 phenolic acids, 46 flavonoids, and 9 anthocyanins were differentially accumulated in the strawberry fruits irradiated with UVB.

5) In the abstract you should write smth about methodology before you write about results. How many light conditions did you test? What kind of light it was? LED light? You didn’t write anything about that.

Response: Thank you for your comments. We are sorry for the misunderstanding caused by our ignorance, in the revised manuscript, we have revised the abstact, please see Lines 11-29:

Lines 11-29: Abstract: Light spectral composition influences the biosynthesis of flavonoids in many plants. However, the detailed composition of flavonoids and anthocyanins and the molecular basis for their biosynthesis in strawberry fruits under two light quality treatments, red light supplemented with blue light (RB) and UVB irradiation remain still unclear. In this stufy, the content of flavonoids and anthocyanins were significantly increased in strawnerry fruits in strawberry fruits under RB light and UVB, respectively. The content of flavonoids and anthocyanins in strawberry fruits under UVB light were dramatically higher than that in strawberry fruits irradiated with RB light. Metabolomic analysis found that a total of 518 metabolites were detected by LC-MS/MS analysis. Among them, 18 phenolic acids, 23 flavonoids, and 8 anthocyanins were differentially accumulated in the strawberry fruits irradiated with red/blue (RB) comparing to other 30 phenolic acids, 46 flavonoids, and 9 anthocyanins fruits irradiated with UVB. The major genes associated with the biosynthesis flavonoids and anthocyanins, including structural genes and transcription factors (TFs), were differentially expressed in the strawberry fruits under RB and UVB irradiation through RNA-seq data analysis. Correlation test of transcriptome and metabolite profiling showed that the expression patterns of most genes in the biosynthesis pathway of flavonoids and anthocyanins were closely correlated with the differential accumulation of flavonoids and anthocyanins. Two TFs, bZIP (FvH4_2g36400) and AP2 (FvH4_1g21210) induced by RB and UVB irradiation, respectively, exhibited similar expression pattern to most structural genes, which was closely correlated with six and eight flavonoids. These results indicated that these two TFs were involved in regulating the biosynthesis of flavonoids and anthocyanins in strawberry furit under RB light and UVB, respectively. These results provide a systematic and comprehensive understanding the accumulation of flavonoids and anthocyanins and the molecular basis for their biosynthesis in strawberry fruits under RB light and UVB.

Introduction:

6) Should be: (Fragaria ×ananassa) (sign × stick to the “ananassa” word).

Response: Thank you for your comments. We are sorry for the misunderstanding caused by our ignorance, in the revised manuscript, we have changed it, please see Line 33:

Line 33: Cultivated strawberry (Fragaria ×ananassa) is perennial herb plant that belongs to Rosaceae family and genus Fragaria, which is a valuable crop culivated worldwide.

  • According to my knowledge names of the botanical families we do not write in italics (Rosaceae).

Response: Thank you for your comments. We are sorry for the misunderstanding caused by our ignorance, in the revised manuscript, we have changed it, please see Line 33:

Line 33: Cultivated strawberry (Fragaria ×ananassa) is perennial herb plant that belongs to Rosaceae family and genus Fragaria, which is a valuable crop culivated worldwide.

  • Is the form of citations correct according to journal requirements? -> Arabic numbers in brackets stick to the last word in the sentence (with no space)?

Response: Thank you for your comments. We are sorry for the misunderstanding caused by our ignorance, in the revised manuscript, we have added a space between Arabic numbers and the last word in the sentence.

  • It should be: “Anthocyanin biosynthesis is also affected by (…)”

Response: Thank you for your comments. We are sorry for the misunderstanding caused by our ignorance, in the revised manuscript, we have revised this sentence, please see Lines 66-67:

Lines 66-67: Anthocyanin biosynthesis is also affected by external environment factors such as temperature, light, drought and so on.

10) I think thise fragment should be used in Discussion section:

Response: Thank you for your comments. We are sorry for the misunderstanding caused by our ignorance, in the revised manuscript, we have revised this sentence, please see Lines 66-67:

“For example, Liu et al. (2018) re-vealed that light quality affects flavonoid production by regulating the related gene ex-pression in Cyclocarya paliurus(15). Blue light could induces biosynthesis of flavonoids in Epimedium sagittatum(16). Kadomura-Ishikawa et al. (2013) has revealed that blue and red light quickly promoted the anthocyanin biosynthesis in fruits by regulating the expression of LAR and ANR genes(17). Illumination with blue light-emitting diodes (LEDs) induced the anthocyanins accumulation in red lettuce(18). Zhang et al.(2018a) has shown that blue and red light increased the contents of total anthocyanins, pelargonidin 3-glucoside, and pelargonidin 3-malonyl glucoside in strawberry fruits(19). Moreover, UV-B can promote flavonoid biosynthesis, leading to improved fruit quality of blueberry(3). Wang et al. (2009) found that the UV radiation increased the anthocyanin contents in blueberry(2).”

  • “The objective of this study was to clarify the effects of red and blue light on strawberry fruit development and maturation.” – how about the UV? You write in the abstract about UV spectrum as well.

Response: Thank you for your comments and suggestions. UVB can promote the accumulat

Materials and methods:

12) ExperimentI – space before I

Response: Thank you for your comments. We are sorry for the misunderstanding caused by our ignorance, in the revised manuscript, we have added a space between Experiment and I, please see Line 118:

Line 118: University. Experiment â… : Strawberry plants with unopened flower buds were selected.

13) Please provide more technical information about LED lighting. What was the spectrum of the used lights in nanometers? What was PPFD? What kind of LED panels they were? How long during the day the supplementary lighting took place? In what time of the year the experiment took place? What other conditions were in the greenhouse (e.g. humidity).

14)  Correct to: (p ≤ 0.05) 

Response: Thank you for your comments. We are sorry for the misunderstanding caused by our ignorance, in the revised manuscript, we have corrected ‘P ≤ 0.05’ into ‘p ≤ 0.05’, please see line 227:

Line 227: The data were analyzed by one-way analysis of variance (ANOVA) and Duncan’s multiple range test (p < 0.05) using SPSS 25.0 statistics.

Results and Discussion:

  • There is no full legend in Figure 1 (you should explain again the abbreviations such as CK, UVB, RB and WF, PF, RF – on chart B shouldn’t be RF instead of FF? And also it should be: ≤ sign in description under Figure.

Response: Thank you for your comments. We are sorry for the misunderstanding caused by our ignorance, in the revised manuscript, we have explained the abbreviations (CK, UVB, and RB), changed FF to RF in Figure 1B, and modify p < 0.05 to p ≤ 0.05, please see Lines 252-262:

Lines 252-262: Figure 1. Morphological traits and physiological indicators under RB- and UVB light. (A) Morphological traits in strawberry fruits irradiated with RB- and UVB light. (B) Effects of RB light on strawberry fruit colaration. Fruits were harvested and photographed in three developmental stages. WF, white fruit; PF, partial red fruit; RF, full red fruit. (C) The contents of flavonoids and anthocyanins. (D) The contents of lignins. (E) The contents of titratable acidity and total sugar. (F) The contents of malondialdehyde (MDA). (G-I) Activities of superoxide dismutase (SOD), catalase (CAT), and peroxidase (POD). The values are means of three replicates. CK, UVB, and RB represent the strawberry fruits irradiated with natural light, Ultraviolet B light and red light supplemented with blue light. Vertical bars indicate the mean standard error. Lowercase letters indicates significant differences between control group and experimental group at p ≤ 0.05 according to Duncan’s multiple range test.

  • Spelling error: strawnerry (Line 4 of 3.1. chapter).

Response: Thank you for your comments. We are sorry for the misunderstanding caused by our ignorance, in the revised manuscript, we have revised it, please see Line 239:

Line 239: The content of flavonoids and anthocyanins were significantly increased in strawberry fruits under RB- and UVB light treatments, respectively.

  • Sentence need to be rewritten: “The content of flavonoids and anthocyanins were significantly increased in strawnerry fruits promote the biosynthesis of flavonoids and anthocyanins in strawberry fruits under RB- and UVB light (Figure 1C).”

Response: Thank you for your comments and suggestions. In the revised manuscript, we have revised this sentence, please see Lines 238-240:

Lines 238-240: The content of flavonoids and anthocyanins were significantly increased in strawberry fruits under RB- and UVB light treatments, respectively.

19) „Table S1), „ – delete space before coma

Response: Thank you for your comments. We are sorry for the misunderstanding caused by our ignorance, in the revised manuscript, we deleted space before coma, please see 263-264:

Lines 263-264: A total of 518 metabolites were identified in strawberry fruits under RB light and UVB light (Table S1), which were divided into four class: flavonoids, phenolic acids, lignins and coumarins, and tannins.

  • Numerious - spelling (Line 4 in Discussion section).

Response: Thank you for your comments. We are sorry for the misunderstanding caused by our ignorance, in the revised manuscript, we revised the spelling and changed into Numerous, please see Line 458:

Line 458: Numerous studies revealed that flavonoid biosynthesis is not only regulated by internal genetic factors but also affected by external factors such as light, temperature and nutritional levels.

  • Comments on the Quality of English Language, Extensive editing of English language required.

Response: Thank you for your comments and suggestions. We are sorry that some small errors in this manuscript. The manuscript was edited for correct English language usage, grammar, punctuation and spelling by qualified native English. Our manuscript has been edited by MDPI's professional English editor.

Reviewer 2 Report

Comments and Suggestions for Authors

Please add references for line 42-46

Line 50: which UV radiation? A,B,C ?

line 62, 66-67:  Required references.

Line 64: remove: etc

M/M section:

Line 136: what ages and height of strawberry seedling was selected? 

please details describe the growing condition and light treatments composition. RH, photoperiod, day/night  temp. nutrient composition and frequency, CO2 level, red/blue light ratios, (UV) light intensity, lighting model and company, natural light intensity, growing medium ( pot, substrate, hydroponic, aeroponic). 

How the 15 fruits were selected, what was the basis/parameters to select fruits to be analyzed? 

please mentioned reference for TF and anthocyanin analysis? the anthocyanin extraction method is not complete or inaccurate. please describe details, calculation methods, with  standard refe. 

174: fine powder

where is the reference for antioxidant enzymes assays ?

3.1: : In results: please mention specific amount where increased or decrease, 

line 543-553: should move to intro section 

553-555: required refe.

556 ;This results were consistent with the previous studies.  where is ref.?

conclusion should be focused on the obtained result with focused  metrics.. please rewrite/modify conclusion.

Comments on the Quality of English Language

 Extensive editing of English language required

Author Response

Dear Reviewer,

On behalf of my co-authors, we thank you very much for giving us an opportunity to revise our manuscript. We have improved our manuscript based on the comments you provided. We appreciate for your warm work earnestly, and hope that the corrections will meet with approval. The responds to the comments are as follows:

Comments and Suggestions for Authors

1)Please add references for line 42-46

Response: Thank you for your comments and suggestions. In the revised manuscript, we have added references for line 42-46, please Lines 42-46”

Lines 42-46: Strawberry fruits are rich in fiber, micronutrient, ascorbic acid, flavonoid and anthocyanins, and offer a broad variety of sensory elicitation and health benefits to the consumer. The color of fruit is an important phase in its life cycle and results from the presence of flavonoid, carotenoid and anthocyanins.

2)Line 50: which UV radiation? A,B,C ?

Response: Thank you for your comments and suggestions. We are sorry for the misunderstanding caused by our ignorance, in the revised manuscript, we revised this sentence, please see Lines 46-47:

Lines 46-47: Anthocyanin content is associated with fruit skin color, and flavonols can shield fruits from UVC radiation (2).

line 62, 66-67: Required references.

Response: Thank you for your comments and suggestions. In the revised manuscript, we have added references for line 55-57. 

3)Line 64: remove: etc

Response: Thank you for your comments and suggestions. We are sorry for the misunderstanding caused by our ignorance, in the revised manuscript, we removed the etc, please see Lines 57-59:

Lines 57-59: Anthocyanin synthesis is not only regulated by internal genetic factors but also affected by external factors such as light, temperature and nutritional levels (5).

M/M section:

4)Line 136: what ages and height of strawberry seedling was selected?

please details describe the growing condition and light treatments composition. RH, photoperiod, day/night  temp. nutrient composition and frequency, CO2 level, red/blue light ratios, (UV) light intensity, lighting model and company, natural light intensity, growing medium ( pot, substrate, hydroponic, aeroponic).

Response: Thank you for your comments and suggestions. We are sorry for the misunderstanding caused by our ignorance, in the revised manuscript, we have added the detail information in “Plants materials and treatments”, please see Lines 121-140:

5)Lines 121-140: The cultivated strawberry (F. ×ananassa ‘Benihoppe’) was used in this study. Two-month-old strawberry seedlings (about 20 cm in height) were cultivated in pots with a volume of 4.5 L, which contained substrate that consisted of a mixture of vermiculite, vermicompost, and soil (2:2:6). The potted strawberry at the 7 days after flowering (7 DAF) were selected and divided into three groups, respectively. The selected strawberry were transfferred in greenhouse with controlled environmental conditions (16 h photoperiod with the light intensity of 100 μmol m−2s−1 at 25 °C, 8 h dark at 16 °C, and 75% relative humidity). Experiment â… : Strawberry plants at 7 DAF were selected and irradiated with the red light supplemented with blue light (RB; red/blue-light photosynthetic photon flux density [PPFD] ratio was 7/1) at 380 mol m−2s−1 PPFD for seven weeks at 25 ℃. Experiment II: Strawberry plants at 7 DAF were selected and irradiated with UVB light for seven weeks at 25 ℃. The strawberry plants at 7 DAF irradiated with natural light were used as the control group (CK).  The RB- and UVB lights (30W, Hangzhou small sun agricultural science and technology co. LTD) ) installed at the top of chambers were applied to treat strawberry seedlings until samples were collected. Strawberry fruits were sampled at 36 (white stage), 43 (pink stage), and 50 (red stage) days after flowering. The collected samples were immediately frozen in liquid nitrogen. The collected strawberry fruits were stored at -80 ℃ for further analyses. Each treatment had three biological replicates and each replicate had 15 fruits from the same stage.

6)How the 15 fruits were selected, what was the basis/parameters to select fruits to be analyzed?

Response: Thank you for your comments and suggestions. The selected parameters are as follows, please see Lines 137-140:

Lines 137-140: Strawberry fruits were sampled at 36 (white stage), 43 (pink stage), and 50 (red stage) days after flowering. The collected samples were immediately frozen in liquid nitrogen. The collected strawberry fruits were stored at -80 ℃ for further analyses. Each treatment had three biological replicates and each replicate had 15 fruits from the same stage.

  • please mentioned reference for TF and anthocyanin analysis? the anthocyanin extraction method is not complete or inaccurate. please describe details, calculation methods, with standard refe.

Response: Thank you for your comments and suggestions. We are sorry for the misunderstanding caused by our ignorance, in the revised manuscript, we have added the reference for TF and anthocyanin analysis and modified the anthocyanin extraction method, please see Lines 150-155.

8)Lines 150-155: Anthocyanins was extracted from strawberry fruit by the procedures that has been described previously [26, 27]. Quantification for total anthocyanins was carried out according to the pH differential method by UV–visible spectrophotometer at 496 nm and 700 nm [28]. The total monomeric anthocyanin concentration was calculated as pelargonidin 3-glucoside, whichwas demonstrated to be the major anthocyanin in strawberry [29].

9)174: fine powder

Response: Thank you for your comments and suggestions. We are sorry for the misunderstanding caused by our ignorance, in the revised manuscript, we have modified this sentence, please see Line 165:

Line 165: Strawberry furits samples (0.1 g) were ground into fine powder in liquid nitrogen and then homogenized with 10 mL of extraction buffer.

10)where is the reference for antioxidant enzymes assays ?

Response: Thank you for your comments and suggestions. We are sorry for the misunderstanding caused by our ignorance, in the revised manuscript, we have added the reference antioxidant enzymes assays, please see lines 171-187:

Lines 171-187:

11)3.1: : In results: please mention specific amount where increased or decrease,

Response: Thank you for your comments and suggestions. We are sorry for the misunderstanding caused by our ignorance, in the revised manuscript, we have added the specific amount where increased or decrease, please see Lines 250-270:

Lines 250-270: As shown in Figure 1, compared to strawberry fruits under natural light, strawberry fruit under supplementary RB- and UVB light showed accelerated color transition during ripening (Figure 1A and 1B). The flavonoid contents were significantly increased in strawberry fruits under RB- and UVB light treatments (385.28 ug/g and 406.59 ug/g, respectively) compared to the control group (57.68 ug/g, Figure 1C). The anthocyanin contents in strawberry fruits irradiated with RB- and UVB lights are 1.15 and 1.67 times that of the control group, respectively (Figure 1C). The anthocyanin contents in strawberry fruits under UVB light were dramatically higher than that in strawberry fruits irradiated with RB light. The lignin content was slightly increased in strawberry fruits treated with RB light (17.78 ug/g) compared to the control group (15.49 ug/g), while UVB light can slighly reduce the lignin content (14.87 ug/g) (Figure 1D). There is no difference in the contents of TA and total sugars between control group and experimental group (RB- and UVB light) (Figure 1E). In addition, the activities of antioxidant enzymes was further investigated. As shown in Figure 1F-I, the contents of POD, CAT, SOD, and MDA were significantly increased in strawberry fruits under RB light treatments (12.76 U/g, 42.99 U/g, 6.74 U/g and 36.59 nmol/g, respectively) compared to the control group (9.83 U/g, 21.69 U/g, 6.03 U/g and 30.46 nmol/g, respectively) The contents of POD, CAT, SOD, and MDA strawberry fruits irradiated with UVB lights are 1.38, 4.14, 1.45, and 1.37 times that of the control group, respectively. The activities of POD, CAT, SOD, and MDA in strawberry fruits irradiated with UVB light was higher than that in strawberry fruits under RB light.

  • line 543-553: should move to intro section

Response: Thank you for your comments and suggestions. In the revised manuscript, we have move to these sentences into introduction, please see Lines 78-85:

13)Lines 78-85:  RB light increased the flavonoid content in A. roxburghii by enhancing the expression of several related genes. Illumination with blue light-emitting diodes (LEDs) induced the anthocyanins accumulation in red lettuce(18). Zhang et al. (2018a) has shown that blue and red light increased the contents of total anthocyanins, pelargonidin 3-glucoside, and pelargonidin 3-malonyl glucoside in strawberry fruits (19). In strawberry, red- and red/blue light caused the earlier fruit coloration by increasing the total anthocyanin contents (33, 34).

14)553-555: required refe.

Response: Thank you for your comments and suggestions. We are sorry for the misunderstanding caused by our ignorance, in the revised manuscript, we have added the reference antioxidant enzymes assays, please see lines 171-187:

15)556 ;This results were consistent with the previous studies.  where is ref.?

Response: Thank you for your comments and suggestions. We are sorry for the misunderstanding caused by our ignorance, in the revised manuscript, we have added the reference antioxidant enzymes assays, please see lines 171-187:

16)conclusion should be focused on the obtained result with focused  metrics.. please rewrite/modify conclusion.

Response: Thank you for your comments and suggestions. We are sorry for the misunderstanding caused by our ignorance, in the revised manuscript, we have modified the conclusion, please see Lines 565-579:

Lines 565-579: In the present study, RB- and UVB light promoted fruit coloration and the accumulation of flavonoids and anthocyanins in strawberry fruits, which also increased the activities of antioxidant enzymes. Metabolomics analysis showed that phenolic acids, flavonoids, and anthocyanins were differentially accumulated in strawberry fruits irradiated with Rb- and UVB light. The essential structure genes and TFs involved in regulating the differential accumulation of flavonoids and anthocyanins in strawberry fruits treated with RB- and UVB light were identified based on RNA-seq data. The correlation analysis between RNA-seq data and metabolite profiling further revealed that two TFs (bZIP, FvH4_2g36400 and AP2, FvH4_1g21210) induced by RB- and UVB irradiation, respectively, exhibited similar expression pattern to most structural genes, which was closely correlated with six and eight flavonoids. These results will deepen our understanding the composition and corresponding molecular mechanisms of flavonoids and anthocyanins biosynthesis in strawberry fruits under RB- and UVB light. Furthermore, the regulatory mechanisms of RB- and UVB on the biosynthesis of flavonoids and anthocyanins in strawberry fruits are different.

17 )Comments on the Quality of English Language

 Extensive editing of English language required.

Response: Our manuscript has been edited by MDPI's professional English editor.

Reviewer 3 Report

Comments and Suggestions for Authors

This study was conducted using strawberries to explore the process of secondary metabolites produced by light quality. Although the light treatment is somewhat simple, the analysis and consideration of the results is logical. It was impressive that they went beyond just observing phenotypes and explored the field of RNA. We believe that the results of this study will greatly contribute to the advancement of this field.

Comments on the Quality of English Language

I believe that the English version of this paper needs minor improvement. We recommend MDPI’s English proofreading service.

Author Response

Dear Reviewer,

On behalf of my co-authors, we thank you very much for giving us an opportunity to revise our manuscript. We have improved our manuscript based on the comments you provided. We appreciate for your warm work earnestly, and hope that the corrections will meet with approval. The responds to the comments are as follows:

 I believe that the English version of this paper needs minor improvement. We recommend MDPI’s English proofreading service.

Response: Our manuscript has been edited by MDPI's professional English editor.

Round 2

Reviewer 2 Report

Comments and Suggestions for Authors

Authors carefully addressed the comments I made. I have gone through entire manuscript, however no issue was detected. 

This manuscript can be proceed further without any correction. 

good luck